# A Preliminary Diagnostic Model for Forward Head Posture among Adolescents Using Forward Neck Tilt Angle and Radiographic Sagittal Alignment Parameters

**DOI:** 10.3390/diagnostics14040394

**Published:** 2024-02-11

**Authors:** Young Jae Moon, Tae Young Ahn, Seung Woo Suh, Kun-Bo Park, Sam Yeol Chang, Do-Kun Yoon, Moo-Sub Kim, Hyeonjoo Kim, Yong Dae Jeon, Jae Hyuk Yang

**Affiliations:** 1Department of Orthopaedic Surgery and Biochemistry, Research Institute of Clinical Medicine of Jeonbuk National University-Biomedical Research Institute of Jeonbuk National University Hospital, Jeonbuk National University Medical School, Jeonju 54896, Republic of Korea; yjmoonos@jbnu.ac.kr; 2Department of Orthopaedic Surgery, Bio-Medical Research Institute, Pusan National University Hospital, Pusan National University School of Medicine, Busan 49241, Republic of Korea; ahnty@pusan.ac.kr; 3Department of Orthopaedics, Scoliosis Research Institute, College of Medicine, Guro Hospital, Korea University, Seoul 08308, Republic of Korea; spine@korea.ac.kr; 4Division of Pediatric Orthopaedic Surgery, Severance Children’s Hospital, Yonsei University College of Medicine, Seoul 03722, Republic of Korea; kunbopark@gmail.com; 5Department of Orthopedic Surgery, College of Medicine, Seoul National University Hospital, Seoul National University, Seoul 03080, Republic of Korea; 6Industrial R&D Center, KAVILAB Co., Ltd., Seoul 06675, Republic of Korea; louis_youn@kavilab.ai (D.-K.Y.); mskim@kavilab.ai (M.-S.K.); hyeonjookim@kavilab.ai (H.K.); yd.jeon84@gmail.com (Y.D.J.); 7Department of Integrative Medicine, College of Medicine, Yonsei University of Korea, Seoul 03722, Republic of Korea; 8Department of Orthopaedics, Scoliosis Research Institute, College of Medicine, Anam Hospital, Korea University, Seoul 02841, Republic of Korea

**Keywords:** cervical spine, forward head posture, forward neck tilt angle, radiological parameters

## Abstract

Despite numerous attempts to correct forward head posture (FHP), definitive evidence-based screening and diagnostic methods remain elusive. This study proposes a preliminary diagnostic methodology for FHP, utilizing a noninvasive body angle measurement system as a screening test for FHP and incorporating radiological parameters for sagittal alignment. We enrolled 145 adolescents for FHP screening. The forward neck tilt angle (FNTA), defined as the angle between the vertical line and the line connecting the participant’s acromion and tragus, was measured using the POM-Checker (a noninvasive depth sensor-based body angle measurement system). A whole-spine standing lateral radiograph was obtained, and eight sagittal alignment parameters were measured. Statistical analyses of the association between the FNTA and eight sagittal alignment parameters were conducted. We used 70% of the participant data to establish a preliminary diagnostic model for FHP based on FNTA and each sagittal alignment parameter. The accuracy of the model was evaluated using the remaining 30% of the participant data. All radiological parameters of sagittal alignment showed weak statistical significance with respect to FNTA (best case: r = 0.16, *p* = 0.0500; cranial tilt). The proposed preliminary diagnostic model for FHP demonstrated 95.35% agreement. Notably, the model using FNTA without radiological parameters accurately identified (100%) participants who required radiographic scanning for FHP diagnosis. Owing to the weak statistical significance of the association between radiological parameters and external body angle, both factors must be considered for accurate FHP diagnosis. When a clear and severe angle variation is observed in an external body angle check, medical professionals should perform radiographic scanning for an accurate FHP diagnosis. In conclusion, FNTA assessment of FNTA through the proposed preliminary diagnostic model is a significant screening factor for selecting participants who must undergo radiographic scanning so that a diagnosis of FHP can be obtained.

## 1. Introduction

Forward head posture (FHP) is an affliction that causes pain in the neck, shoulders, and chest caused by maintaining a downward head angle to stare at, for example, smartphones, personal computers, or books for a long time. Good posture allows the head to weigh ~10–14 lb in the neutral supine position. However, a head tilt of 60° increases the weight-bearing capacity of the head to 60 lb. Specifically, smartphones are accelerating the occurrence of FHP, regardless of age.

Smartphones have been universally accepted as information and communication devices and have become indispensable in modern everyday life. Smartphone use has increased rapidly, especially among young adults and adolescents, and is often causing health-related problems [1]. Regarding the musculoskeletal system, neck discomfort or pain associated with an FHP is of the greatest concern. In prospective longitudinal studies, several authors have also observed that the duration of smartphone use is associated with musculoskeletal symptoms such as neck pain [2,3]. People naturally maintain their necks in a flexed position when using a smartphone and thus exhibit an FHP, as confirmed by laboratory studies [4,5]. FHP leads to increased loads imparted to the cervical spine [6], causing adverse outcomes in the musculoskeletal system. Clinically, the results from a longitudinal study by Okada et al. also support the idea that sagittal alignment of the cervical spine impacts intervertebral disc degeneration [7]. A school-based epidemiological study of 1196 adolescents identified that FHP or decreased craniovertebral angle (CVA) measured using digital photographs with retroreflective markers were associated with increased odds of a lifetime prevalence of neck pain [8].

Considering the clinical significance of FHP, as evidenced in the literature, proper screening and early detection of FHP are crucial for adolescents in this smartphone era. However, limited evidence exists regarding an appropriate FHP screening method, which should be non-invasive and straightforward to apply. Furthermore, studies that validate these noninvasive screening methods by comparing them to radiographic measurements as the gold standard for sagittal alignment assessment are scarce in the literature. Although the diagnosis of FHP is replaced by the observation of symptoms, such as headaches, neck discomfort, muscle tension in the neck and shoulders, discomfort in the midback, and chest pain, the above symptoms can disturb the correct diagnosis of diseases other than FHP. External posture is also a key observation point regarding FHP. However, there are no clear criteria for using external posture to diagnose FHP. Moreover, some recent studies have reported finding no significant correlations between radiographic parameters and FHP diagnosis. Therefore, it is difficult to establish a diagnostic model for FHP using only radiographic images. Moreover, although external posture can indicate an FHP, radiographic parameters can show different results, leading to misdiagnosis. Furthermore, unnecessary scans, including those involving radiation exposure, are often thought to be required to diagnose FHP. However, the diagnosis of FHP by only checking external posture also carries a sufficiently high risk of misdiagnosis.

There are several considerations required to accurately diagnose FHP. First, the definition of the external posture is essential and must be quantified. Second, the process and results of the quantification of external posture should be consistent and reliable, and the quantity of data should be sufficient to obtain statistical insight. Third, clear relationships between the quantified external posture and radiographic parameters should be statistically investigated. The last consideration entails the establishment of a diagnostic model based on a clear relationship between quantified external posture and radiographic parameters. This model can be employed to reliably diagnose FHP and can provide criteria allowing the avoidance of unnecessary radiographic scanning for diagnosing FHP. In this study, the external posture of participants was quantified by measuring the forward neck tilt angle (FNTA) using a measurement device. Eight radiographic parameters were measured from the X-ray images for the sagittal alignment of the same participant.

The purpose of this study was to suggest a preliminary diagnostic methodology for FHP based on a noninvasive body angle measurement system for measuring FNTA as a screening test for FHP and radiological parameters for sagittal alignment and to evaluate the accuracy of the preliminary diagnostic model. This study introduces a preliminary diagnostic model for FHP among adolescents, combining the use of a noninvasive body angle measurement system, specifically assessing the FNTA, with a detailed radiographic analysis. This preliminary model enhances existing diagnostic approaches by providing a more comprehensive assessment of FHP, including its potential impact on the cervical spine. In particular, the approach outlined in this study offers a significant advancement over current models by integrating both noninvasive and radiographic techniques. This dual methodology not only allows for preliminary noninvasive screening but also ensures detailed diagnostic accuracy through radiographic evaluation. This is particularly crucial in the adolescent population, among whom the early detection and accurate diagnosis of FHP are essential for the effective intervention in and prevention of long-term complications.

## 2. Materials and Methods

### 2.1. Ethics Approval and Consent to Participate

Adolescent volunteers (ages 10–19) who agreed to be screened for FHP, regardless of the presence or severity of related symptoms, were recruited from five different institutions. A total of 145 participants were selected from among adolescents with no history of tumors, infections, fractures, or previous surgery of the spine and who were able to maintain an upright posture during the required examinations. The current study was approved by the institutional review board of the respective institutions (IRB file no: Jeonbuk National University Hospital; 2020-12-046, Pusan National University Hospital; H-2011-025-096, Seoul National University Hospital; H-2011-035-1170, Korea University Guro Hospital; 2020GR0539, Yonsei University Severance Hospital; 4-2021-0125), and all participants and their guardians provided informed consent.

### 2.2. Forward Neck Tilt Angle Measurement Using a Depth-Based Body Angle Measurement System

The FNTA, defined as the angle between a vertical line and the line connecting the participant’s shoulder (acromion) and ear (tragus), was measured using a depth-sensor-based body angle measurement system. POM-Checker (Team Elysium Inc., Seoul, Republic of Korea) is a non-invasive, markerless, depth-sensor-based system that can measure various body angles and motions using Kinect-V2 (Microsoft, Seattle, WA, USA) [9]. FNTA was measured in a standardized standing posture, and the participants were instructed to stand upright with their knees locked, eyes looking straight ahead, elbows bent, and knuckles positioned in the supraclavicular fossa bilaterally, as recommended in the radiographic measurement manual of the spinal deformity study group (SDSG) [10]. For standardized measurements, the participants stood 2 m away from the POM-Checker, which was installed 1.5 m above the floor. Figure 1a shows an example of the FNTA measurement process. After the measurement of the FNTA, the FHP levels of the participants were classified according to the FNTAs: Normal (0° ≤ FNTA < 12°), Level 1 (12° ≤ FNTA < 25°), Level 2 (25° ≤ FNTA < 40°), and Level 3 (40° ≤ FNTA). According to our definition, ‘Level 3’ is a level requiring radiographic imaging in order to diagnose the FHP. This classification was previously used at the FHP level in the present study.

### 2.3. Radiographic Measurements 

Whole-spine lateral radiographs were obtained using a standardized protocol. The participants were instructed to stand as recommended in the SDSG manual, and radiographs were obtained using two 14 × 17-inch films in a 14 × 36 cassette. The following three parameters were measured from the digital X-ray images using a picture-archiving and communication system: cervical sagittal alignment (CSA), distance between the plumb line dropped from the centroid of C2, and the posterior superior aspect of C7. The craniocervical angle (CCA) was measured between the line from the hard palate to the opisthion and the line from the hard palate to the center of C7. T1 slope (T1S): This angle was measured between the horizontal plane and a line parallel to the T1 endplate. Cervical lordosis (CL; C2–C7 Cobb angle) corresponds to the angle between a line parallel to the inferior endplate of C2 and that parallel to the superior endplate of C7. Cranial tilt (CranT) corresponds to the angle between the line to the dens from the center of C7 and the sagittal vertical axis from C7. Cervical tilt (CervT) corresponds tothe angle formed between the vertical line from the center of C7 and the line from the center of C7 to the tip of the dens. C7 slope (C7S) corresponds to the angle between the horizontal line and upper endplate of C7. Thoracic kyphosis (TK; T4–T12 Cobb angle) corresponds to the Cobb angle between the superior endplate of T4 and the inferior endplate of T12. Figure 1b shows the angle definitions using the axes for the eight radiographic parameters [11,12].

### 2.4. Statistical Analysis 

First, the normality for all continuous variables, including the FNTA and eight radiographic parameters, was investigated using the Shapiro–Wilk normality test in MATLAB (2023a, Mathworks, Natick, MA, USA). Generally, to investigate the normality of a sample, two representative tests are employed: the Shapiro–Wilk normality test and the Kolmogorov–Smirnov test. The Kolmogorov–Smirnov test is an effective test procedure when there are thousands of sample data. In other words, the test model is suitable for large quantities of data. In contrast, the Shapiro–Wilk normality test is more suitable for a small quantity of data, like the dataset used in this study. Variables were considered normally distributed when the *p*-value exceeded 0.05. The Spearman’s rho correlation is a nonparametric measure suitable for monotonic relationships. It can handle ordinal data and is less sensitive to outliers, whereas Pearson’s correlation is appropriate for linear relationships between continuous variables, assuming there is a normal distribution. If there were any correlations between the FNTA and the eight radiographic parameters, there would be a strong connecting link with which to decide the FHP level using the FNTA. However, several studies have failed to show a clear connecting link between FNTA and radiographic parameters. For this reason, there have been no clear criteria with which to diagnose FHP using externally measured angles and radiographic parameters. In this study, we progressed the statistical analysis in relation to two aspects. The first hypothesis was that when the sample data had ranked variables, the data would follow an ordinal tendency. In this study, the FHP level is the ranked variable. The second hypothesis was that the sample data would simply follow a linear correlation. The FNTA has linearity. When the data for radiographic parameters were also linear, a strong connecting link would be found. The Spearman rho correlation and Pearson’s correlation are suitable models for validating the first and second hypotheses, respectively. For this reason, we considered two correlation models: Spearman’s Rho and Pearson’s correlation. Moreover, we investigated the mean and standard deviation (SD) of each radiographic parameter according to FHP levels.

### 2.5. Establishment of a Preliminary Diagnostic Model for FHP 

A reasonable diagnostic model can assist in making diagnostic decisions due to its use of multiple parameters. The diagnostic model in this study provides a decision on the FHP level using the FNTA and radiographic parameters. At this stage, the classification of the FHP level had already been conducted according to the FNTA, and the FHP level can be easily classified using only the FNTA. However, because the diagnosis of FHP using only FNTA is associated with a high probability of misdiagnosis, radiographic parameters should be included in a diagnostic model. Furthermore, when the significance between the FNTA and each radiographic parameter is high, the classification line between two groups in a scatter plot according to groups can be a reasonable classification model. The preliminary diagnostic model was established using multivariate analysis. In this case, the linear equation defining the boundary line between groups (group 1 and 2) was derived as 0 = K + (L1 × X) + (L2 × Y). It can be transformed into Y = −(K + (L1 × X))/L2. Y is a component of the boundary line, and X represents the variables in the radiographic parameter, where K, L1, and L2 are constants determined by the data distribution in the scatter plot. If only FNTA is employed to classify the level, then a perfect horizontal line classifies the level. However, this linear equation will have various slopes owing to the influence of radiographic parameters. In this study, four groups of three boundary lines were used. In particular, the classification line between Levels 2 and 3 is important because it can provide information useful for the request for radiology. Scatter plotting was performed using 102 participants (70% of the total participant data) who were randomly extracted with consideration of balance according to the FHP level. Because there were eight radiographic parameters, eight diagnostic models for FHP were generated according to each radiographic parameter.

### 2.6. Accuracy Evaluation for Diagnostic Model 

Accuracy evaluation of the FHP diagnostic model is an important process that indicates the usefulness of the preliminary diagnostic model for FHP. For accuracy evaluation, we employed the remaining data on 43 participants (accounting for 30% of the total participant data), constituting the data that remained after the establishment of the diagnostic model. The diagnosis of FHP for accuracy evaluation was performed using only FNTA without radiographic parameters. When the decision regarding FHP level was difficult to conclude because of ambiguous FNTA, both levels were considered candidates for the FHP level. The eventual diagnosis of FHP was determined by selecting the candidate level that received the most votes from all eight diagnostic models. Figure 2 shows the overall workflow of this study. 

## 3. Results

### 3.1. Statistical Results

Table 1 shows the results of the normality test, the Spearman’s Rho, and Pearson’s correlation between the FNTA and each radiographic parameter and between the FHP level and each radiographic parameter. The highest significance, 0.16 (*p* = 0.0500), was reported for the relationship between FNTA and C7S. However, most relationships were not strongly correlated. Table 2 shows the means and SDs of the measured parameters, including FNTA, according to the FHP level. There are no specific regularities with which to distinguish the levels, except for the FNTA. However, dramatic variations were observed in CCA, CranT, and CervT at Level 3, which is a level requiring radiology for an FHP diagnosis. 

### 3.2. Preliminary Diagnostic Model for FHP

Based on these eight radiographic parameters, eight individual diagnostic models were used to diagnose FHP. Each model had three lines for classifying the groups, which were defined by a linear equation. Figure 3 and Figure 4 show the preliminary diagnostic models for FHP. All models show the measured angle (FNTA) and each radiographic parameter on the X- and Y-axes, respectively. The sky-blue circles, blue upside-down triangles, pink triangles, and red crosses show the distributions for Normal and Levels 1, 2, and 3. The blue line distinguishes between Normal and Level 1. The pink line represents the classification line between Levels 1 and 2. The red line represents the boundary between levels 2 and 3. Table 3 lists the constants (K, L1, and L2) used to define the three classification lines for each diagnostic model. Moreover, the left/right endpoints for the X-axis and the initial/endpoint for the Y-axis are also listed to limit the graph according to the radiographic parameters.

### 3.3. Accuracy of the Preliminary Diagnostic Model

The preliminary diagnostic model for FHP was established by defining three boundary lines, which can be divided into two groups using the scatter plot. To evaluate the accuracy of the preliminary diagnostic model for FHP, only the FNTA data of 43 participants were used to diagnose FHP levels. Table 4 shows the overall result sheet showing the performance of the diagnostic model for assessing FHP levels. The first and second columns present the number of participants and their FNTAs, respectively. Numbers 1–8 in the third to tenth columns show the candidate level(s) from the eight diagnostic models based on eight radiographic parameters. Numbers 1–8 indicate CSA, CCA, T1S, CL, CranT, CervT, C7S, and TK, respectively, in regular sequences. N, L1, L2, and L3 are the abbreviations for Normal and Levels 1 to 3, respectively, and they represent the counted number of candidate levels according to the FHP level from all the diagnostic models. The penultimate column (PC) was previously classified as the FHP level according to FNTA, which is irrelevant to radiographic parameters. The final FHP level selected from the candidate levels is shown in the last column (D: Diagnosis). The agreement ratio between the previously classified FHP level and the diagnosis was 95.35%. Figure 5 shows a confusion chart using the penultimate column and last column in Table 4. The X- and Y-axes indicate the diagnosis using eight diagnostic models and the previously classified FHP level when using only FNTA. The true-positive rate (TPR) is the ratio of correctly identified positive instances to the total number of actual positive instances. The false-negative rate (FNR) is the ratio of incorrectly identified negative instances to the total number of positive instances. The positive predictive value (PPV) is the ratio of correctly identified positive instances to the total predicted positive instances. The false discovery rate (FDR) is the ratio of incorrectly identified positive instances to the total number of predicted positive instances.

## 4. Discussion

Modern technologies are reshaping everyday life and posture. Smartphones often bend our necks and induce postures with various terminologies, including iPosture, text-neck, computer neck, turtle neck, Dowager’s hump, and FHP. Many researchers have investigated the relationship between an abnormal posture, smartphone use, and neck pain or discomfort among adolescents. In a large-scale school-based epidemiological study, Dolphens et al. reported that FHP was significantly associated with an increased lifetime prevalence of neck pain and the need for medical help for neck pain [8]. In contrast, some authors found no statistically significant associations between FHP and neck pain in this population [13,14,15]. There seems to be a relationship between temporomandibular joint (TMJ) disorders and forward head posture, which may be a significant consideration because the incidence of TMJ disorders in the female population is much higher than that in the male population [16,17].

The CVA was the most commonly used metric for measuring FHP in previous studies [18]. In these studies, the CVA, defined as the angle between the horizontal line and the line connecting the participant’s ear (tragus) and the C7 spinous process tip, was measured using reflective markers. Although some authors consider CVA > 50° to be pathological FHP [19], no consensus has been reached on the definition or diagnostic criteria for FHP in the literature. Evidence of a correlation between CVA and radiological measurements of sagittal cervical alignment is scarce. In this study, we measured the FNTA, which is defined as the angle between the vertical line and the line connecting a participant’s ear (tragus) and shoulder (acromion). FNTA was measured using a noninvasive, depth-sensor-based body angle measurement system, POM-Checker. Measurement using the POM-Checker is fast, straightforward, and biologically harmless because it does not require marker attachment, a change of clothing, or any kind of stimulated emission of radiation exposure. Therefore, it may be a more suitable screening method for FHP than conventional anthropometric measurements using reflective markers.

Although a simple way to diagnose FHP can be the use of only FNTA, failing to consider radiographic parameters can hamper a correct diagnosis. Moreover, the current study showed a statistically weak correlation between FNTA and each radiographic parameter. The weak statistical significance between the FNTA and radiological parameters can be interpreted as an indication that the two groups do not have a significant relationship. Nevertheless, our interpretation is that there is a more complicated relationship between FNTA and each radiographic parameter. Regardless of the interpretation, when a diagnosis of FHP is conducted, if only one factor is considered, it will be difficult for a diagnosis to achieve reliability. Therefore, the diagnosis of FHP should consider both FNTA and each radiographic parameter. The perfect exclusion of radiographic scanning to diagnose FHP is not feasible in the current stage. However, radiographic scanning cannot be performed on all patients to diagnose FHP. Thus, a proper compromise is required for a reasonable diagnosis. This study suggested a preliminary diagnostic methodology for FHP assessment based on a noninvasive body angle measurement system for measuring FNTA as a screening test for FHP and radiological parameters for sagittal alignment. Ultimately, we focused on the diagnostic method using only FNTA and found a weak correlation with radiographic parameters. A simple classification based on the previously classified FHP level uses an absolute horizontal line to classify the level. In other words, the FHP level can be determined via comparison with the acceptable range for each level. However, our proposed model has three boundary lines with different slopes, which are defined by the scatter between the radiographic parameters and FNTA. As the proposed model is composed of data from 102 participants, each slope can change slightly when the data are added. Generally, there have been many machine learning studies that use a general ratio of data allocation, such as 7(Training): 3(Test) and 6(Training): 2(Validation): 2(Test). Naturally, there is no correct answer regarding data allocation; rather, researchers are finding the proper allocation ratio through repeated training. This study is in the pre-stage with respect to applying machine learning methods, and it was possible to skip the validation dataset. We had to confirm the results when the machine learning algorithm trained the same data. For this reason, we allocated 70% of the participant data to establish the preliminary diagnostic model and allocated the remaining 30% for evaluation. 

Simple classification according to the previously classified FHP level requires radiology when the FNTA is >40°. However, the proposed model can perform radiographic scanning using a previous radiographic reference. The effective range for the diagnosis of level 3 in this study was 40–50°. This type of statistical approach increases the accuracy and reliability of FHP diagnosis. As a result, the diagnostic accuracy of the preliminary diagnostic model in this study was reported to be 95.35% using 43 patient data. In other words, there is a difference between FNTA-based classification and diagnosis using the FNTA and radiographic parameters. The value of this difference, 4.65%, is a factor that can affect the diagnosis of FHP using the radiographic parameters. Moreover, it can be sufficiently different according to the number of data, and this is a factor that cannot be ignored. From the results of this study, we can draw several clinically significant points. First, when a robust diagnostic model for FHP is established using more reliable data, the diagnosis of FHP no longer requires radiologic scanning. Naturally, the current stage is an initial stage before proceeding to the next step, which is to abolish radiologic scanning for the diagnosis of FHP. However, the current model can at least select the patients who must be radiologically scanned to allow for a diagnosis of FHP. This means that the average radiation exposure will be lower for patients as well as health care providers. Second, the results of this study can be employed to diagnose FHP at institutes that do not have sufficient patient data for FHP. We collected 145 adolescent patient data relevant to FHP, and the collection took a long time. If a medical doctor can measure FNTA using mobile devices such as a smartphone or a tablet PC, like the case for the POM-Checker, the results of this study can be employed as a temporary diagnostic model until sufficient data are collected. Third, the results of this study can explain and depict the process of using machine learning for the diagnosis of FHP. Using machine learning after training the same data in this study provided the same predictions as the results in Figure 5. Because the purpose of machine learning in this regard is to find the coefficients K, L1, and L2 in Table 3, if we know those coefficients, we can easily decide the class in a manner similar to machine learning. However, it is not easy to visualize the process of machine learning. In particular, when multiple complicated variables are required to decide something, like in the case of this study, visualization is more difficult. For this reason, when machine learning is trained on the investigators’ clinical experience data, an understanding of the process of machine learning will be helpful in establishing a reliable model.

The current study has several limitations. First, we included participants aged 10–19. This age range can be regarded as wide, considering that the sagittal alignment of the spine changes as a child grows and the amount of smartphone usage varies significantly with age [20,21,22,23,24,25,26,27]. For the same reason, the age of the participants has been suggested to be the most critical confounding factor in related studies [13]. Second, the standardized standing position for the FNTA measurement in our study differed from the participants’ postures in everyday life, especially when using smartphones. Third, the relationship between smartphone usage and FHP was not analyzed because of the characteristics of the participants. In the future, we will need to upgrade the preliminary diagnostic model using more investigations and data. Lastly, all adolescents participating in this study underwent radiography. Naturally, conducting radiography for research can be problematic. This study only targeted people who met the following conditions: (1) people who had not been subjected to radiological imaging for the purposes of examination or treatment within one year; (2) people who were suspected of having a disease such as turtle neck (FHP) on the outside but could not be tested; people who wanted a clear radiology test for FHP; and (3) people who had been given a sufficient explanation of the biological problems that radiography may cause and had given consent. Explanations and consent for all of the above items were provided to each child and guardian who participated. Moreover, only few images (one lateral spine image taken while standing) were taken for the X-ray radiography of voluntary participants who met the three conditions mentioned above and sought a diagnosis of FHP. By progressing this study, an FHP screening test method will be developed to minimize unnecessary radiological examinations for growing children in the future, and guidelines for radiographic imaging will be developed. This study was conducted because radiography is considered necessary for the validity of studies in this field. 

Sarig Bahat et al. (2023) investigated the relationship between FHP and neck pain, employing craniovertebral angle measurements and specialized virtual reality software [28]. Their findings indicated that there were no significant correlations between FHP and neck pain, offering a contrasting perspective to our approach, which combines noninvasive measurements with radiographic data. Chu et al. (2020) explored the impact of FHP on upper cervical spine stability [29]. This study utilized radiographic analysis to assess changes in joint spacing related to FHP, offering insights into its potential effects on cervical spine stability. In contrast, our research focuses on a diagnostic model that integrates both noninvasive body angle measurements and detailed radiographic parameters. Our study stands out by providing a comprehensive diagnostic model for FHP and integrating both noninvasive body angle measurements and detailed radiographic parameters. 

The current study was based on statistical analysis and did not employ a machine learning approach. Machine learning can be used to solve these issues. Specifically, a diagnosis of FHP can be easily made by training Naïve Bayes, ensemble, and decision tree algorithms. Additionally, we can select the best model by simultaneously training several models. However, the training data also require labeled data for the diagnosis level of FHP. To use machine learning to diagnose FHP, sufficient diagnostic criteria for the diagnostic model should be established prior to the application of machine learning. This study can serve as a starting point for the application of machine learning in the diagnosis of FHP.

## 5. Conclusions

FNTA has been highlighted as a fast and biologically harmless screening method for FHP, with advantages over conventional anthropometric measurements. Owing to the limited statistical significance observed between radiological parameters and external body angle, it is imperative to consider both factors for an accurate diagnosis of FHP. In instances where distinct angle variations are evident during an assessment of the external body angle (FNTA), medical professionals should proceed with radiographic scanning to ensure precision in diagnosing FHP. In conclusion, the evaluation of FNTA using the proposed preliminary diagnostic model can effectively and reliably diagnose FHP levels without radiographic scanning. The proposed model can be a crucial screening criterion aiding the identification of participants warranting radiographic scanning for FHP diagnosis.

## Figures and Tables

**Figure 1 diagnostics-14-00394-f001:**
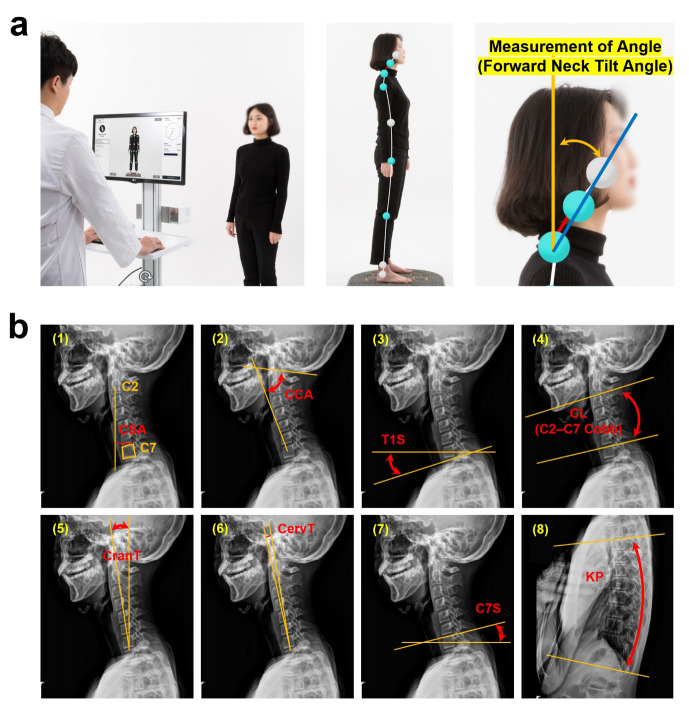
The measurement process for the forward neck tilt angle using the POM-Checker and radiographic parameters form the X-ray image for sagittal alignment. (**a**) Assessment of forward neck tilt angle (FNTA; angle between blue and orange line) followed a standardized standing position protocol. Participants were directed to maintain an erect stance, with their knees fully extended, gaze fixed straight ahead, and knuckles positioned in the supraclavicular fossa on both sides. This aligns with the guidelines outlined in the radiographic measurement manual of the spinal deformity study group (SDSG). (**b**) Eight radiographic parameters: (1) cervical sagittal alignment (CSA, C2–C7), (2) craniocervical angle (CCA), (3) T1 slope (T1S), (4) cervical lordosis (CL; C2–C7 Cobb angle), (5) cranial tilt (CranT), (6) cervical tilt (CervT), (7) C7 slope (C7S), and (8) thoracic kyphosis (TK; T4–T12).

**Figure 2 diagnostics-14-00394-f002:**
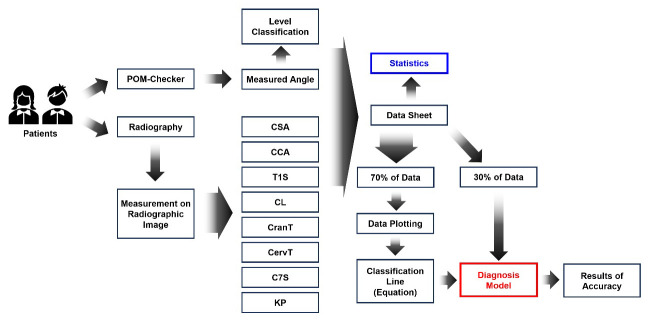
Overall workflow of the process of establishing the preliminary diagnostic model for forward head posture (FHP). The measurement of forward neck tilt angle (FNTA) for all participants was carried out using POM-Checker, a non-invasive depth sensor-based system for assessing body angles in a standardized standing position. Subsequently, a full-spine standing lateral radiograph was acquired, and the measurement of eight sagittal alignment parameters ensued. A statistical examination (blue box) comparing FNTA with the eight sagittal alignment parameters was conducted. Furthermore, a preliminary diagnostic model for FHP, based on both FNTA and individual sagittal alignment parameters, was created using 70% of participant data. The accuracy of the model (red box) was then assessed using the remaining 30% of participant data.

**Figure 3 diagnostics-14-00394-f003:**
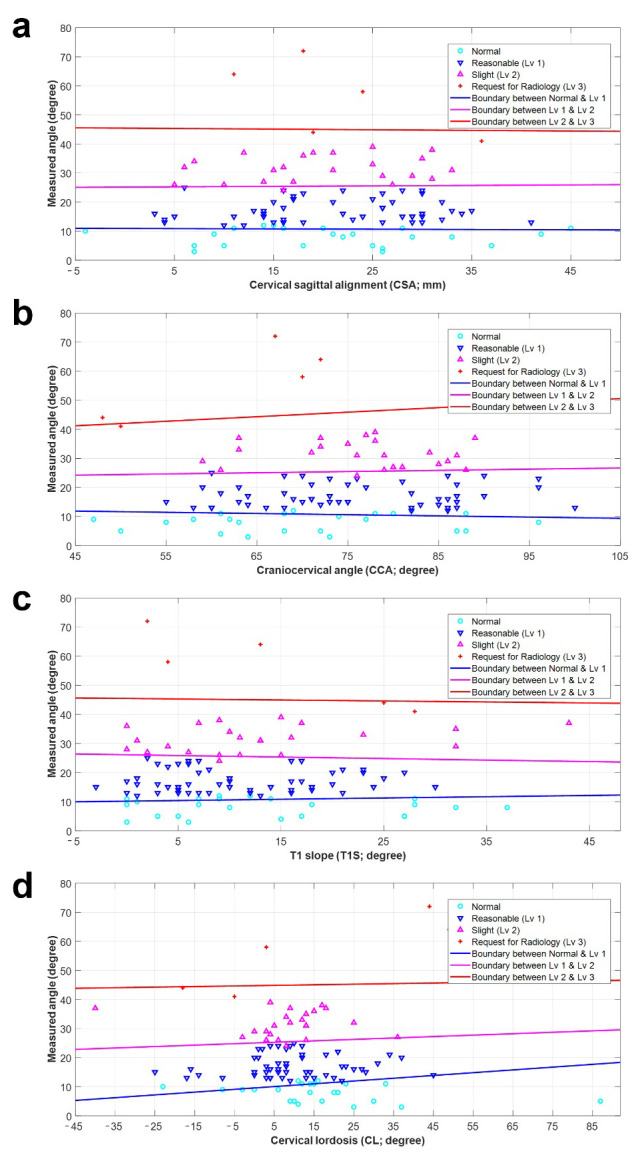
Preliminary diagnostics model for forward head posture according to radiographic parameters: (**a**) cervical sagittal alignment (CSA, C2–C7), (**b**) craniocervical angle (CCA), (**c**) T1 slope (T1S), and (**d**) cervical lordosis (CL; C2–C7 Cobb angle). Each model illustrates the angle measurement (forward neck tilt angle) and individual radiographic parameters on the X-axis and Y-axis, respectively. Representing the distributions for Normal, Level 1, Level 2, and Level 3 are sky-blue circles, blue upside-down triangles, pink triangles, and red crosses, respectively. A delineating blue line was established to distinguish between Normal and Level 1. The pink line serves as the classification boundary between Level 1 and Level 2, while the red line marks the boundary between Level 2 and Level 3.

**Figure 4 diagnostics-14-00394-f004:**
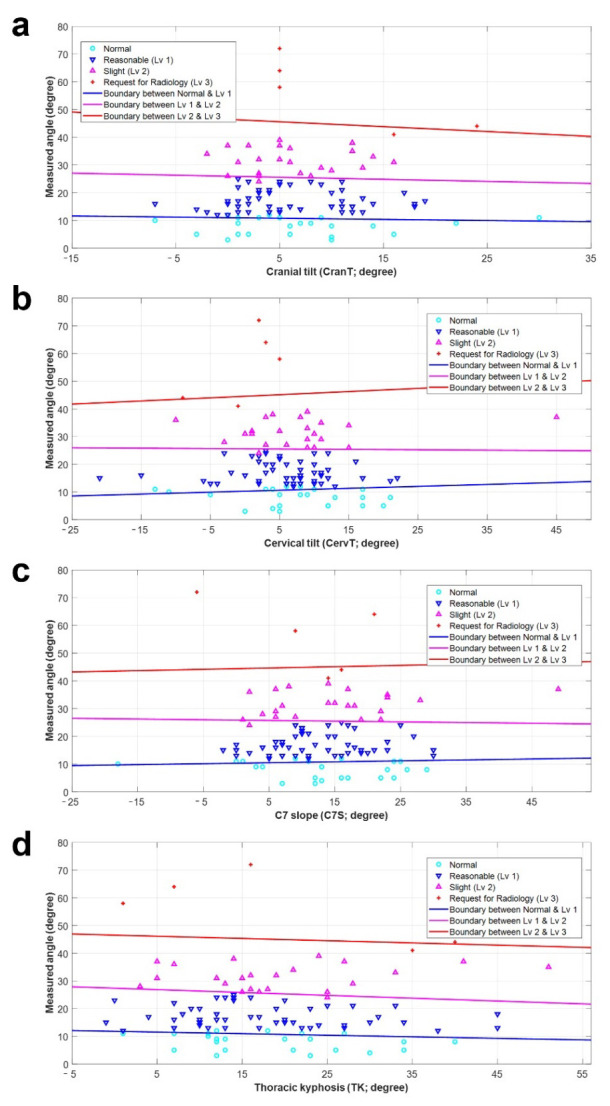
Preliminary diagnostics model for forward head posture according to radiographic parameters: (**a**) cranial tilt (CranT), (**b**) cervical tilt (CervT), (**c**) C7 slope (C7S), and (**d**) thoracic kyphosis (TK; T4-T12). All models present the angle measurement (forward neck tilt angle) and specific radiographic parameters along the X-axis and Y-axis, respectively. The distributions for Normal, Level 1, Level 2, and Level 3 are represented by sky-blue circles, blue upside-down triangles, pink triangles, and red crosses, respectively. An outlining blue line was introduced to differentiate between Normal and Level 1. The pink line functions as the classification boundary separating Level 1 and Level 2, whereas the red line delineates the boundary between Level 2 and Level 3.

**Figure 5 diagnostics-14-00394-f005:**
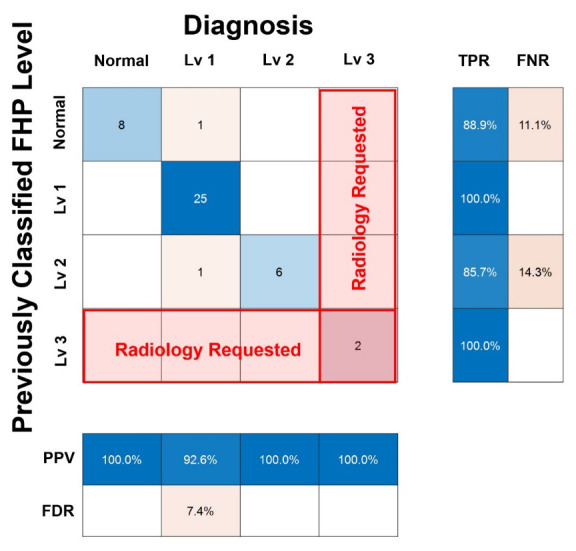
Confusion chart of diagnosis made using eight diagnostic models, and the previously classified FHP level. The X-axis and Y-axis represent the outcomes obtained from the application of eight diagnostic models and the singular forward neck tilt angle (FNTA)-based FHP level classification, respectively. The model accurately classified the Level 3 which is a level for radiology requested (red boxes). The true-positive rate (TPR) is defined as the proportion of accurately identified positive instances relative to the total actual positive instances. Conversely, the false-negative rate (FNR) is characterized as the proportion of erroneously identified negative instances in relation to the overall actual positive instances. Positive predictive value (PPV) signifies the proportion of correctly identified positive instances among the total predicted positive instances, while the false discovery rate (FDR) is the proportion of inaccurately identified positive instances with respect to the total number of predicted positive instances.

**Table 1 diagnostics-14-00394-t001:** Results of correlation analysis between the forward neck tilt angle (FNTA) and each radiographic parameter and between forward head posture level (FHP Lv) and each radiographic parameter. The variable distribution follows a normal distribution, and the Spearman’s Rho and Pearson’s correlation were investigated. The eight radiographic parameters are cervical sagittal alignment (CSA, C2–C7), craniocervical angle (CCA), T1 slope (T1S), cervical lordosis (CL; C2–C7 Cobb angle), cranial tilt (CranT), cervical tilt (CervT), C7 slope (C7S), and thoracic kyphosis (TK; T4–T12).

	N * (*p*)	S * (FNTA)	*p*	P * (FNTA)	*p*	S * (FHP Lv)	*p*	P * (FHP Lv)	*p*
CSA	0.6031	0.00	0.9781	0.00	0.9779	−0.06	0.4921	−0.02	0.7805
CCA	0.6798	−0.02	0.8047	−0.06	0.4660	−0.08	0.3223	−0.10	0.2256
T1S	0.1063	0.11	0.1753	0.07	0.4306	0.04	0.6702	0.02	0.8409
CL	0.0638	0.13	0.1200	0.02	0.8125	0.07	0.4219	−0.01	0.9435
CranT	0.0848	0.02	0.8215	0.01	0.9335	0.03	0.7216	0.01	0.8637
CervT	0.1379	−0.11	0.1702	0.01	0.9479	−0.11	0.2065	−0.01	0.9375
C7S	0.9129	0.16	0.0500	0.14	0.0890	0.11	0.1711	0.12	0.1584
TK	0.2849	−0.11	0.1801	−0.04	0.6374	−0.09	0.2582	−0.05	0.5832

* N = Normality, * S = Spearman, and * P = Pearson.

**Table 2 diagnostics-14-00394-t002:** Means and standard deviations of the measured parameters, including the forward neck tilt angle (FNTA), according to forward head posture (FHP) level. The FHP levels for participants were classified according to the FNTA: Normal (0° ≤ FNTA < 12°), Level 1 (12° ≤ FNTA < 25°), Level 2 (25° ≤ FNTA < 40°), and Level 3 (40° ≤ FNTA). The abbreviations of radiographic parameters are listed as follows: cervical sagittal alignment (CSA, C2–C7), craniocervical angle (CCA), T1 slope (T1S), cervical lordosis (CL; C2–C7 Cobb angle), cranial tilt (CranT), cervical tilt (CervT), C7 slope (C7S), and thoracic kyphosis (TK; T4–T12). The FHP levels are Normal, Level 1, Level 2, and Level 3 (request for radiology).

	N *	L1 *	L2 *	L3 * †
Age	13.91 ± 2.04	13.93 ± 2.17	13.45 ± 1.68	13.14 ± 2.12
CSA	19.61 ± 10.62	19.20 ± 10.38	19.62 ± 7.76	22.14 ± 9.75
CCA	71.64 ± 12.18	76.25 ± 11.19	76.55 ± 8.50	64.86 ± 13.55
T1S	12.88 ± 10.12	10.67 ± 8.09	13.76 ± 10.51	13.00 ± 11.37
CL	14.18 ± 17.67	10.11 ± 13.46	8.07 ± 13.44	16.71 ± 26.32
CranT	6.03 ± 7.51	5.76 ± 5.82	5.97 ± 4.72	12.29 ± 7.48
CervT	7.64 ± 7.88	6.08 ± 7.75	7.76 ± 9.22	3.43 ± 7.57
C7S	14.03 ± 9.93	12.72 ± 7.97	16.03 ± 10.58	14.71 ± 12.65
TK	18.45 ± 10.16	16.59 ± 10.21	20.72 ± 10.74	17.43 ± 15.02
FNTA	8.61 ± 5.04	17.25 ± 3.82	31.24 ± 4.36	54.71 ± 12.34

* N: Normal level, * L1: FHP level 1, * L2: FHP level 2, and * L3: FHP level 3, †: Request for radiology.

**Table 3 diagnostics-14-00394-t003:** List of constants (K, L1, and L2) used to define the three classification lines for each diagnostic model. The blue, pink, and red lines are boundary lines between Normal and Level 1, between Level 1 and Level 2, and between Level 2 and Level 3, respectively. The left/right endpoints for the X-axis and the initial/end point for Y-axis are also listed. For reference, the FHP levels for participants were classified according to the FNTA: Normal (0° ≤ FNTA < 12°), Level 1 (12° ≤ FNTA < 25°), Level 2 (25° ≤ FNTA < 40°), and Level 3 (40° ≤ FNTA).

	Color	K	L1	L2	LE	RE	IA	EA
CSA	Blue(B)	4.7059	−0.0041	−0.4309	−5	50	11.0	10.4
	Pink(P)	17.0615	0.0112	−0.6780	−5	50	25.1	26.0
	Red(R)	52.0386	−0.0249	−1.1452	−5	50	45.5	44.4
CCA	B	5.8290	−0.0175	−0.4254	45	105	11.9	9.4
	P	15.3719	0.0281	−0.6869	45	105	24.2	26.7
	R	41.0884	0.1879	−1.2028	45	105	41.2	50.6
T1S	B	4.3672	0.0187	−0.4290	−5	48	10.0	12.3
	P	17.8022	−0.0355	−0.6815	−5	48	26.4	23.6
	R	52.2012	−0.0393	−1.1483	−5	48	45.6	43.8
CL	B	4.3264	0.0432	−0.4527	−45	92	5.30	18.3
	P	17.4075	0.0341	−0.6955	−45	92	22.8	29.5
	R	51.7508	0.0232	−1.1564	−45	92	43.8	46.6
CranT	B	4.7627	−0.0175	−0.4333	−15	35	11.6	9.6
	P	17.7856	−0.0508	−0.6855	−15	35	27.1	23.4
	R	54.5616	−0.2062	−1.1743	−15	35	49.1	40.3
CervT	B	4.4493	0.0301	−0.4330	−25	50	8.5	13.8
	P	17.3439	−0.0097	−0.6775	−25	50	26.0	24.9
	R	51.4267	0.1307	−1.1542	−25	50	41.7	50.2
C7S	B	4.4401	0.0147	−0.4315	−25	54	9.4	12.1
	P	17.5074	−0.0172	−0.6774	−25	54	26.5	24.5
	R	50.9228	0.0547	−1.1472	−25	54	43.2	47.0
TK	B	5.2067	−0.0249	−0.4419	−5	56	12.1	8.6
	P	19.4449	−0.0732	−0.7108	−5	56	27.9	21.6
	R	55.2256	−0.0951	−1.1870	−5	56	46.9	42.0

Color: Line Color, LE: Left Angle, RE: Right Angle, IA: Initial Angle, and EA: End Angle.

**Table 4 diagnostics-14-00394-t004:** Comprehensive result sheet demonstrating the diagnostic model’s performance for FHP levels. The first and second columns showcase participant numbers and their corresponding FNTA values. Columns three to ten present numerical assignments (1 to 8) representing candidate levels derived from eight diagnostic models based on distinct radiographic parameters (CSA, CCA, T1S, CL, CranT, CervT, C7S, and TK). The abbreviations N, L1, L2, and L3 indicate Normal and Levels 1 to 3, respectively, and signify the tally of candidate levels across all diagnostic models corresponding to FHP levels. The penultimate column, labeled PC, denotes the previously classified FHP level by FNTA, which is entirely unrelated to radiographic parameters. The last column reveals the ultimately selected FHP level from the candidate levels (D: Diagnosis). For reference, the FHP levels for participants were classified according to the FNTA: Normal (0° ≤ FNTA < 12°), Level 1 (12° ≤ FNTA < 25°), Level 2 (25° ≤ FNTA < 40°), and Level 3 (40° ≤ FNTA).

	FNTA	1	2	3	4	5	6	7	8	N	L1	L2	L3	PC	D
1	13	1	1	1	0.1	1	0.1	1	1	2	8	0	0	1	1
2	15	1	1	1	1	1	1	1	1	0	8	0	0	1	1
3	17	1	1	1	1	1	1	1	1	0	8	0	0	1	1
4	17	1	1	1	1	1	1	1	1	0	8	0	0	1	1
5	12	1	1	0.1	0.1	1	0.1	0.1	0.1	5	8	0	0	1	1
6	21	1	1	1	1	1	1	1	1	0	8	0	0	1	1
7	21	1	1	1	1	1	1	1	1	0	8	0	0	1	1
8	11	1	0.1	0.1	0.1	0.1	0.1	0.1	0.1	7	8	0	0	0	1
9	33	2	2	2	2	2	2	2	2	0	0	8	0	2	2
10	31	2	2	2	2	2	2	2	2	0	0	8	0	2	2
11	23	1	1	1	1	1	1	1	1.2	0	8	1	0	1	1
12	17	1	1	1	1	1	1	1	1	7	8	0	0	1	1
13	10	0	0.1	0	0.1	0.1	0.1	0.1	0.1	8	6	0	0	0	0
14	23	1	1	1	1	1	1	1	1.2	0	8	1	0	1	1
15	15	1	1	1	1	1	1	1	1	0	8	0	0	1	1
16	36	2	2	2	2	2	2	2	2	0	0	8	0	2	2
17	23	1	1	1	1	1	1	1	1.2	0	8	1	0	1	1
18	29	2	2	2	2	2	2	2	2	0	0	8	0	2	2
19	6	0	0	0	0	0	0	0	0	8	0	0	0	0	0
20	16	1	1	1	1	1	1	1	1	0	8	0	0	1	1
21	13	1	1	1	0.1	1	0.1	1	1	2	8	0	0	1	1
22	4	0	0	0	0	0	0	0	0	8	0	0	0	0	0
23	17	1	1	1	1	1	1	1	1	0	8	0	0	1	1
24	1	0	0	0	0	0	0	0	0	8	0	0	0	0	0
25	25	1	1.2	1.2	1.2	1.2	1.2	1.2	1.2	0	8	7	0	2	1
26	27	2	2	2	1.2	1.2	2	2	1.2	0	3	8	0	2	2
27	31	2	2	2	2	2	2	2	2	0	0	8	0	2	2
28	12	1	1	0.1	0.1	1	0.1	0.1	0.1	5	8	0	0	1	1
29	52	3	3	3	3	3	3	3	3	0	0	0	8	3	3
30	13	1	1	1	0.1	1	0.1	1	1	2	8	0	0	1	1
31	20	1	1	1	1	1	1	1	1	0	8	0	0	1	1
32	7	0	0	0	0.1	0	0	0	0	8	1	0	0	0	0
33	15	1	1	1	1	1	1	1	1	0	8	0	0	1	1
34	19	1	1	1	1	1	1	1	1	0	8	0	0	1	1
35	9	0	0	0	0.1	0	0.1	0	0.1	8	3	0	0	0	0
36	13	1	1	1	0.1	1	0.1	1	1	2	8	0	0	1	1
37	19	1	1	1	1	1	1	1	1	0	8	0	0	1	1
38	21	1	1	1	1	1	1	1	1	0	8	0	0	1	1
39	21	1	1	1	1	1	1	1	1	0	8	0	0	1	1
40	22	1	1	1	1	1	1	1	1.2	0	8	1	0	1	1
41	10	0	0.1	0	0.1	0.1	0.1	0.1	0.1	8	6	0	0	0	0
42	62	3	3	3	3	3	3	3	3	0	0	0	8	3	3
43	10	0	0.1	0	0.1	0.1	0.1	0.1	0.1	8	6	0	0	0	0

## Data Availability

We have deposited the sample code we authored on GitHub (https://github.com/Louis-Youn/Code_Storage.git, accessed on 10 September 2023) to facilitate exploration by the wider community. The sample code with the employed dataset shared via GitHub reflects the materials used in this study. Because the full datasets are still protected by privacy issues and regulation policies, an inquiry regarding access to the full dataset can be made by contacting the corresponding author (J.H.Y., E-mail: kuspine@korea.ac.kr).

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
