# Peer review of "A Preliminary Diagnostic Model for Forward Head Posture among Adolescents Using Forward Neck Tilt Angle and Radiographic Sagittal Alignment Parameters"

_diagnostics, 2024, doi:10.3390/diagnostics14040394_

Round 1
Reviewer 1 Report
Comments and Suggestions for Authors
Dear Editor,
Thank you for your invitation to review and congratulations to the authors for their work.
I have some suggestions for improving the paper and they are given below.
In the first occurrence of abbreviations, please write the open form of the words. For example, the explanation of the term "FNTA" is given in the abstract but not in the article.
In this sense, I suggest you review the whole article.
I think the most important problem for this study is that children were evaluated with x-rays. Because the children received radiation for the study. Please explain if this was a routine practice or if the children's existing x-rays were used. However, the ethics committee approved this study.
Please explain if all children were x-rayed.
In Table 1, the test names in the top column make the table complicated. For this reason, I recommend writing the test names at the bottom of the table. You can use the * symbol to indicate which test you are using for which assessment.
Explain the definitions of level 1 and level 2. These statements in the table are not understood.
In general, I suggest that all tables should be made clearer and more understandable so that readers can understand them. In the discussion section, it would be valuable to emphasize the clinical significance of this research and the clinical experience of the investigators.
Author Response
Reviewer 1
Thank you for your invitation to review and congratulations to the authors for their work. I have some suggestions for improving the paper and they are given below.
[Authors’ response] Thank you for consuming your precious time to improve the quality of our manuscript with your great comments. And we really appreciate to bring the positive aspects of our study to us. According to your comments, our manuscript has been revised. We would like to ask you to review our responses for your comments.
- In the first occurrence of abbreviations, please write the open form of the words. For example, the explanation of the term "FNTA" is given in the abstract but not in the article. In this sense, I suggest you review the whole article.
[Authors’ response] Thank you for your significant remark. Actually, we have found the wrong use of the abbreviation rule for several words except for your example. We have added the full name at the first use of the word according to the abbreviation rule. The below list shows a list of the modified abbreviations in the text.
FNTA [Modified Words, Page: 2, Line: 96]
SDSG [Modified Words, Page: 3, Line: 136]
PC [Modified Words, Page: 10, Line: 324]
TMJ [Modified Words, Page: 13, Line: 370]
CVA [Modified Words, Page: 13, Line: 374]
- I think the most important problem for this study is that children were evaluated with x-rays. Because the children received radiation for the study. Please explain if this was a routine practice or if the children's existing x-rays were used. However, the ethics committee approved this study. Please explain if all children were x-rayed.
[Authors’ response] Really appreciate. All children participating in the study underwent radiography. Of course, doing radiography for research can be problematic.
Accordingly, our researchers only targeted people who met the following conditions: 1) People who had not taken radiological images for the purpose of examination or treatment within one year; 2) People who were suspected of having a disease such as turtle neck (FHP) on the outside but could not be tested; People who want a clear radiology test for FHP, 3) People who have heard a sufficient explanation about the biological problems that radiography may cause and have given consent. In addition, explanation and consent for all of the above items were provided to each target child and guardian. In addition, only minimal images (one lateral spine image taken while standing) were taken for X-ray radiography of voluntary participants who met the three conditions mentioned above and sought a diagnosis of FHP.
The impact of one plain X-ray radiography on the human body is minimal. In addition, among the various radiographs taken to check the condition of the spine, side imaging of the spine has a smaller effect than anterior and posterior imaging. Also, please consider that the study volunteers wanted to diagnose FHP, and radiography was performed for diagnostic purposes.
Additionally, the purpose of this study was to determine whether the apparent cervical kyphotic deformity identified in the patient's clinical photographs was actually related to the kyphotic deformity of the cervical bones identified in radiography. Through this, if there is a correlation between the deformation on clinical photographs and the kyphotic deformity of the cervical spine confirmed radiologically, the FHP screening test method will be developed to minimize unnecessary radiological examinations for growing children in the future, and guidelines for radiographic imaging will be developed. This study was conducted because radiography was necessary for the validity of the study.
[Added Contents, Page: 14-15, Line: 459-473]
Lastly, all adolescents participating in the study underwent radiography. Naturally, doing radiography for research can be problematic. This study only targeted people who met the following conditions: 1) People who had not taken radiological images for the purpose of examination or treatment within one year; 2) People who were suspected of having a disease such as turtle neck (FHP) on the outside but could not be tested; People who want a clear radiology test for FHP, 3) People who have heard a sufficient explanation about the biological problems that radiography may cause and have given consent. And explanation and consent for all of the above items were provided to each target child and guardian. Moreover, only minimal images (one lateral spine image taken while standing) were taken for X-ray radiography of voluntary participants who met the three conditions mentioned above and sought a diagnosis of FHP. By progressing this study, the FHP screening test method will be developed to minimize unnecessary radiological examinations for growing children in the future, and guidelines for radiographic imaging will be developed. This study was conducted because radiography was necessary for the validity of the study.
- In Table 1, the test names in the top column make the table complicated. For this reason, I recommend writing the test names at the bottom of the table. You can use the * symbol to indicate which test you are using for which assessment.
[Authors’ response] Thank you for your great comment on elevating the intuition of the table. According to your comment, we have changed the table. Especially, the full test names have been located at the bottom of the table as the annotation.
[Modified Table, Page: 7, Line: 262]
|
|
N* (p) |
S* (FNTA) |
p |
P* (FNTA) |
p |
S* (FHP Lv) |
p |
P* (FHP Lv) |
p |
|
CSA |
0.6031 |
0.00 |
0.9781 |
0.00 |
0.9779 |
-0.06 |
0.4921 |
-0.02 |
0.7805 |
|
CCA |
0.6798 |
-0.02 |
0.8047 |
-0.06 |
0.4660 |
-0.08 |
0.3223 |
-0.10 |
0.2256 |
|
T1S |
0.1063 |
0.11 |
0.1753 |
0.07 |
0.4306 |
0.04 |
0.6702 |
0.02 |
0.8409 |
|
CL |
0.0638 |
0.13 |
0.1200 |
0.02 |
0.8125 |
0.07 |
0.4219 |
-0.01 |
0.9435 |
|
CranT |
0.0848 |
0.02 |
0.8215 |
0.01 |
0.9335 |
0.03 |
0.7216 |
0.01 |
0.8637 |
|
CervT |
0.1379 |
-0.11 |
0.1702 |
0.01 |
0.9479 |
-0.11 |
0.2065 |
-0.01 |
0.9375 |
|
C7S |
0.9129 |
0.16 |
0.0500 |
0.14 |
0.0890 |
0.11 |
0.1711 |
0.12 |
0.1584 |
|
TK |
0.2849 |
-0.11 |
0.1801 |
-0.04 |
0.6374 |
-0.09 |
0.2582 |
-0.05 |
0.5832 |
*N = Normality, *S = Spearman, *P = Pearson
- Explain the definitions of level 1 and level 2. These statements in the table are not understood.
[Authors’ response] We are sorry to cause confusion by the insufficient description. Although the definition of the level of FHP was described in the 2.2 Section in Material and Method (Page 3, Line 129 – 130; After the measurement of the FNTA, the FHP levels for participants were classified according to the FNTA: Normal (0° ≤ FNTA < 12°), Level 1 (12° ≤ FNTA < 25°), Level 2 (25° ≤ FNTA < 40°), Level 3 (40° ≤ FNTA).), we believe the readability was too low to recognize the definition. To improve the readability of the definition of the FHP level, we have added the description to the table caption which requires an understanding of the FHP level.
[Modified Table Caption, Page: 7, Line: 263-270]
Table 2. Results of mean and standard deviation of the measured parameter including the for-ward neck tilt angle (FNTA) according to forward head posture (FHP) level. The FHP levels for participants were classified according to the FNTA: Normal (0° ≤ FNTA < 12°), Level 1 (12° ≤ FNTA < 25°), Level 2 (25° ≤ FNTA < 40°), Level 3 (40° ≤ FNTA). The abbreviations of radiographic parameters are listed as follows: cervical sagittal alignment (CSA, C2–C7), craniocervical angle (CCA), T1 slope (T1S), cervical lordosis (CL; C2–C7 Cobb angle), cranial tilt (CranT), cervical tilt (CervT), C7 slope (C7S), thoracic kyphosis (TK; T4–T12). The FHP level includes Normal, Level 1, Level 2, and Level 3 (request for radiology).
[Modified Table Caption, Page: 10, Line: 306-311]
Table 3. List of constants (K, L1, and L2) to define the three classification lines for each diagnostic model. The blue, pink, and red lines are boundary lines between Normal and Level 1, between Level 1 and Level 2, and between Level 2 and Level 3, respectively. The left/right endpoints for the X-axis and the initial/end point for Y-axis were also listed. For reference, the FHP levels for participants were classified according to the FNTA: Normal (0° ≤ FNTA < 12°), Level 1 (12° ≤ FNTA < 25°), Level 2 (25° ≤ FNTA < 40°), Level 3 (40° ≤ FNTA).
[Modified Table Caption, Page: 11, Line: 338-348]
Table 4. Comprehensive result sheet, demonstrating the diagnostic model's performance for FHP levels. The first and second columns showcase participant numbers and their corresponding FNTA values. Columns three to ten present numerical assignments (1 to 8) representing candidate levels derived from eight diagnostic models based on distinct radiographic parameters (CSA, CCA, T1S, CL, CranT, CervT, C7S, and TK). The abbreviations N, L1, L2, and L3 indicate Normal, Level 1 to 3, respectively, and signify the tally of candidate levels across all diagnostic models corresponding to FHP levels. The penultimate column, labeled PC, denotes the previously classified FHP level by FNTA, which is entirely unrelated to radiographic parameters. The last column reveals the ultimately selected FHP level from the candidate levels (D: Diagnosis). For reference, the FHP levels for participants were classified according to the FNTA: Normal (0° ≤ FNTA < 12°), Level 1 (12° ≤ FNTA < 25°), Level 2 (25° ≤ FNTA < 40°), Level 3 (40° ≤ FNTA).
- In general, I suggest that all tables should be made clearer and more understandable so that readers can understand them. In the discussion section, it would be valuable to emphasize the clinical significance of this research and the clinical experience of the investigators.
[Authors’ response] Thank you for your good suggestions and comments. To make the table easily understandable, we have modified the complicated tables. Especially, Table 1 currently has annotations to present the test name. Table 2 also has annotations to express the FHP levels. The variables in the top line of Table 3 have been modified as the abbreviation with the annotations. Moreover, we have added the clinical significance of this research and clinical experience of the investigators.
Table 1 [Modified Table, Page: 7, Line: 262]
|
|
N* (p) |
S* (FNTA) |
p |
P* (FNTA) |
p |
S* (FHP Lv) |
p |
P* (FHP Lv) |
p |
|
CSA |
0.6031 |
0.00 |
0.9781 |
0.00 |
0.9779 |
-0.06 |
0.4921 |
-0.02 |
0.7805 |
|
CCA |
0.6798 |
-0.02 |
0.8047 |
-0.06 |
0.4660 |
-0.08 |
0.3223 |
-0.10 |
0.2256 |
|
T1S |
0.1063 |
0.11 |
0.1753 |
0.07 |
0.4306 |
0.04 |
0.6702 |
0.02 |
0.8409 |
|
CL |
0.0638 |
0.13 |
0.1200 |
0.02 |
0.8125 |
0.07 |
0.4219 |
-0.01 |
0.9435 |
|
CranT |
0.0848 |
0.02 |
0.8215 |
0.01 |
0.9335 |
0.03 |
0.7216 |
0.01 |
0.8637 |
|
CervT |
0.1379 |
-0.11 |
0.1702 |
0.01 |
0.9479 |
-0.11 |
0.2065 |
-0.01 |
0.9375 |
|
C7S |
0.9129 |
0.16 |
0.0500 |
0.14 |
0.0890 |
0.11 |
0.1711 |
0.12 |
0.1584 |
|
TK |
0.2849 |
-0.11 |
0.1801 |
-0.04 |
0.6374 |
-0.09 |
0.2582 |
-0.05 |
0.5832 |
N* = Normality, S* = Spearman, P* = Pearson
Table 2 [Modified Table, Page: 7, Line: 271]
|
|
N* |
L1* |
L2* |
L3* † |
|
Age |
13.91 ± 2.04 |
13.93 ± 2.17 |
13.45 ± 1.68 |
13.14 ± 2.12 |
|
CSA |
19.61 ± 10.62 |
19.20 ± 10.38 |
19.62 ± 7.76 |
22.14 ± 9.75 |
|
CCA |
71.64 ± 12.18 |
76.25 ± 11.19 |
76.55 ± 8.50 |
64.86 ± 13.55 |
|
T1S |
12.88 ± 10.12 |
10.67 ± 8.09 |
13.76 ± 10.51 |
13.00 ± 11.37 |
|
CL |
14.18 ± 17.67 |
10.11 ± 13.46 |
8.07 ± 13.44 |
16.71 ± 26.32 |
|
CranT |
6.03 ± 7.51 |
5.76 ± 5.82 |
5.97 ± 4.72 |
12.29 ± 7.48 |
|
CervT |
7.64 ± 7.88 |
6.08 ± 7.75 |
7.76 ± 9.22 |
3.43 ± 7.57 |
|
C7S |
14.03 ± 9.93 |
12.72 ± 7.97 |
16.03 ± 10.58 |
14.71 ± 12.65 |
|
TK |
18.45 ± 10.16 |
16.59 ± 10.21 |
20.72 ± 10.74 |
17.43 ± 15.02 |
|
FNTA |
8.61 ± 5.04 |
17.25 ± 3.82 |
31.24 ± 4.36 |
54.71 ± 12.34 |
*N: Normal level, *L1: FHP level 1, *L2: FHP level 2, *L3: FHP level 3, †:Request for radiology
Table 3 [Modified Table, Page: 10, Line: 312]
|
|
Color |
K |
L1 |
L2 |
LE |
RE |
IA |
EA |
|
CSA |
Blue(B) |
4.7059 |
-0.0041 |
-0.4309 |
-5 |
50 |
11.0 |
10.4 |
|
Pink(P) |
17.0615 |
0.0112 |
-0.6780 |
-5 |
50 |
25.1 |
26.0 |
|
|
Red(R) |
52.0386 |
-0.0249 |
-1.1452 |
-5 |
50 |
45.5 |
44.4 |
|
|
CCA |
B |
5.8290 |
-0.0175 |
-0.4254 |
45 |
105 |
11.9 |
9.4 |
|
P |
15.3719 |
0.0281 |
-0.6869 |
45 |
105 |
24.2 |
26.7 |
|
|
R |
41.0884 |
0.1879 |
-1.2028 |
45 |
105 |
41.2 |
50.6 |
|
|
T1S |
B |
4.3672 |
0.0187 |
-0.4290 |
-5 |
48 |
10.0 |
12.3 |
|
P |
17.8022 |
-0.0355 |
-0.6815 |
-5 |
48 |
26.4 |
23.6 |
|
|
R |
52.2012 |
-0.0393 |
-1.1483 |
-5 |
48 |
45.6 |
43.8 |
|
|
CL |
B |
4.3264 |
0.0432 |
-0.4527 |
-45 |
92 |
5.30 |
18.3 |
|
P |
17.4075 |
0.0341 |
-0.6955 |
-45 |
92 |
22.8 |
29.5 |
|
|
R |
51.7508 |
0.0232 |
-1.1564 |
-45 |
92 |
43.8 |
46.6 |
|
|
CranT |
B |
4.7627 |
-0.0175 |
-0.4333 |
-15 |
35 |
11.6 |
9.6 |
|
P |
17.7856 |
-0.0508 |
-0.6855 |
-15 |
35 |
27.1 |
23.4 |
|
|
R |
54.5616 |
-0.2062 |
-1.1743 |
-15 |
35 |
49.1 |
40.3 |
|
|
CervT |
B |
4.4493 |
0.0301 |
-0.4330 |
-25 |
50 |
8.5 |
13.8 |
|
P |
17.3439 |
-0.0097 |
-0.6775 |
-25 |
50 |
26.0 |
24.9 |
|
|
R |
51.4267 |
0.1307 |
-1.1542 |
-25 |
50 |
41.7 |
50.2 |
|
|
C7S |
B |
4.4401 |
0.0147 |
-0.4315 |
-25 |
54 |
9.4 |
12.1 |
|
P |
17.5074 |
-0.0172 |
-0.6774 |
-25 |
54 |
26.5 |
24.5 |
|
|
R |
50.9228 |
0.0547 |
-1.1472 |
-25 |
54 |
43.2 |
47.0 |
|
|
TK |
B |
5.2067 |
-0.0249 |
-0.4419 |
-5 |
56 |
12.1 |
8.6 |
|
P |
19.4449 |
-0.0732 |
-0.7108 |
-5 |
56 |
27.9 |
21.6 |
|
|
R |
55.2256 |
-0.0951 |
-1.1870 |
-5 |
56 |
46.9 |
42.0 |
Color: Line Color, LE: Left Angle, RE: Right Angle, IA: Initial Angle, EA: End Angle
[Added Contents, Page: 14, Line: 427-449]
From the results of this study, we could draw several clinically significant points. First, when the robust diagnostic model for the FHP is established by more reliable data, the diagnosis of the FHP does not require radiologic scanning anymore. Naturally, the current stage is an initial stage to enter the next step which is to remove the radiologic scanning for the diagnosis of the FHP. However, at least the current model can select the patients who need radiologic scanning for the diagnosis of the FHP. It means that the average radiation exposure will be essentially decreased for the patients as well as health care providers. Second, the results of this study can be employed to diagnose the FHP at institutes that do not have sufficient patient data for the FHP. We collected 145 adolescent patient data relevant to FHP, and the collection took a long time. If the medical doctor can measure the FNTA using mobile devices such as a smartphone, or a tablet PC, like the POM-Checker, the results of this study can be employed as a temporary diagnostic model until sufficient data is collected. Third, the results of this study can explain and visualize the process of machine learning for the diagnosis of FHP. The machine learning after training the same data in this study can provide the same predictions as the results in Figure 5. Because the purpose of machine learning is to find the coefficients K, L1, and L2 in Table 3, if we know those coefficients, we can easily decide the class like machine learning. However, it is not easy to visualize the process of machine learning. Especially, when complicated multiple variables are required to decide something, like case of this study, the visualization is more difficult. For this reason, when machine learning trains the investigators’ clinical experience data, the understanding of the process of machine learning will be helpful in establishing a reliable model.
Reviewer 2 Report
Comments and Suggestions for Authors
Dear author,
Find the attached comments and do the necessary changes wherever applicable.

Author Response
Reviewer 2
- I appreciate the effort you've put into investigating a diagnostic methodology for forward head posture (FHP). Your study presents a valuable contribution to the field, and the proposed preliminary diagnostic model is a promising step toward addressing the challenges associated with FHP diagnosis. I would like to commend you on the clarity of your methodology and the use of a non-invasive body angle measurement system in conjunction with radiological parameters. The combination of the forward neck tilt angle (FNTA) and sagittal alignment parameters provides a comprehensive approach to FHP screening. However, as with any scientific study, I have a few suggestions for minor revisions that I believe would enhance the overall quality and clarity of your manuscript.
[Authors’ response] Thank you for consuming your precious time to improve the quality of our manuscript with your great comments. And we really appreciate to bring the positive aspects of our study to us. According to your comments, our manuscript has been revised. We would like to ask you to review our responses for your comments.
- Clarification of Statistical Analyses: Provide more details on the statistical analyses conducted between the FNTA and the eight sagittal alignment parameters. Specify the statistical tests used and consider reporting effect sizes.
[Authors’ response] Thank you for your good comments. As your comments, we have added more detail on the statistical analyses conducted between the FNTA and the eight sagittal alignment parameters, and we have specified the statistical test used. However, we originally did not consider the effect size. Because the FHP level is a classified and ranked variable, it is hard to calculate the effect size. The effect size between FNTA and radiographic parameters can be calculated. The below table shows the results of the effect size based on Cohen’s d. As you can see, the various range was shown. This is a natural result. First, each measurement indicator is individual and has a different range. For this reason, we would like to find the correlation relationship. Second, the weak correlations from Spearman rho and Pearson’s correlation were presented. Although the two sample groups have similar mean, we were very cautious to define the relation with weak correlation and similarity of the mean. Third, because we previously expected a weak correlation by several references, the research design was constructed to investigate the FNTA-based diagnosis with a low impact on radiographic parameters. This is why we did not consider the effect size between the FNTA and radiographic parameters.
|
|
CSA |
CCA |
T1S |
CL |
CranT |
CervT |
C7S |
TK |
|
FNTA |
-0.0340 |
4.7088 |
-0.7484 |
-0.6516 |
-1.4421 |
-1.3014 |
-0.5732 |
-0.1781 |
[Added Contents, Page: 5, Line: 176-181]
Generally, to investigate the normality of the sample, two representative tests are employed as the Shapiro–Wilk normality test and the Kolmogorov-Smirnov test. In the case of the Kolmogorov-Smirnov test, it is an effective test procedure when there are thousands of sample data. It means that the test model is proper for large amounts of data. Whereas, the Shapiro–Wilk normality test is more proper for a small amount of data like the data in this study.
[Added Contents, Page: 5, Line: 185-197]
If there are any correlations between the FNTA and eight radiographic parameters, a strong connecting link will be available to decide the FHP level using the FNTA. However, a lot of studies have failed to show a clear connecting link between FNTA and radiographic parameters. For this reason, there have been no clear criteria to diagnose FHP using externally measured angles and radiographic parameters. In this study, we progressed the statistical analysis as two aspects. The first hypothesis is that when the sample data has ranked variables, the data will follow the ordinal tendency. In this study, the FHP level is the ranked variable. The second hypothesis is that sample data will simply follow the linear correlation. The FNTA has a linearity. When the data for radiographic parameters has also linearity, a strong connecting link will be found. The Spearman rho correlation and Pearson’s correlation are proper models to validate the first hypothesis and second hypothesis, respectively.
- Discussion of Weak Statistical Significance: In the discussion, elaborate on the implications of the weak statistical significance observed between FNTA and radiological parameters. Address any potential limitations or confounding factors that may contribute to this finding.
[Authors’ response] Thank you. As your comment, we have added the description to elaborate on the implications of the weak statistical significance observed between FNTA and radiological parameters. Moreover, we have addressed potential limitations and confounding factors.
[Modified Contents, Page: 13, Line: 388-398]
Although a simple way to diagnose FHP can be the use of only FNTA, not considering radiographic parameters can disturb the correct diagnosis. Moreover, the current study showed a statistically weak correlation between FNTA and each radiographic parameter. The weak statistical significance between the FNTA and radiological parameters can be interpreted as the two groups do not have actual pertinent. Nevertheless, our interpretation is that there is a more complicated relationship between FNTA and each radiographic parameter. Regardless of interpretation, when the diagnosis of FHP is conducted, if only one factor is considered, it is true that the diagnosis can be hard to reach reliability. Therefore, the diagnosis of FHP should consider both FNTA and each radiographic parameter, the perfect exclusion of radiographic scanning to diagnose FHP is not proper in the current stage.
- Validation of Preliminary Diagnostic Model: Discuss the methodology and rationale behind using 70% of participant data to establish the preliminary diagnostic model and the remaining 30% for evaluation. Additionally, provide insights into the implications of the 95.35% agreement observed
[Authors’ response] Thank you for your precious comments to improve the quality of the manuscript. Generally, there have been many machine learning studies that use a general ratio of data allocation as 7:3, 6:2:2. Naturally, there is no correct answer about data allocation, rather than the researchers are finding the proper allocation ratio through several training. This study is in the pre-stage to apply the machine learning methodology and it was possible to skip the validation dataset. We have to confirm the results when the machine learning trained the same data. For this reason, we allocated 70% of participant data to establish the preliminary diagnostic model and the remaining 30% for evaluation. Lastly, the diagnostic accuracy of the preliminary diagnostic model was reported as 95.35% using 43 patient data. According to your comment, we have added the discussion points and insight.
[Added Contents, Page: 13, Line: 409-417]
Generally, there have been many machine learning studies that use a general ratio of data allocation as 7(Training):3(Test), 6(Training):2(Validation):2(Test). Naturally, there is no correct answer about data allocation, rather than the researchers are finding the proper allocation ratio through several training. This study is in the pre-stage to apply the machine learning methodology and it was possible to skip the validation dataset. We have to confirm the results when the machine learning trained the same data. For this reason, we allocated 70% of participant data to establish the preliminary diagnostic model and the remaining 30% for evaluation.
[Added Contents, Page: 14, Line: 422-427]
As a result, the diagnostic accuracy of the preliminary diagnostic model in this study was reported as 95.35% using 43 patient data. In other words, there is a difference between FNTA-based classification and the diagnosis using the FNTA and radiographic parameters. The value of this 4.65% is a probability that can affect the diagnosis of FHP by the radiographic parameters. Moreover, it can be sufficiently different by the amount of the data, and this is a probability that cannot be ignored.
- Consideration of Ethical and Practical Implications: Discuss any ethical considerations or practical implications related to the decision making process for radiographic scanning based on external body angle variations. This could include considerations for minimizing unnecessary radiographic exposure. Overall, your study is well-conducted and presents a valuable contribution to the understanding of FHP diagnosis. I believe these minor revisions will further enhance the clarity and impact of your manuscript.
[Authors’ response] We really appreciate your compliment and great comment on our study. We agree with your concern about ethical considerations or practical implications. We have addressed the discussion for ethical considerations and practical implications as the one of limitation.
[Added Contents, Page: 14, Line: 459-473]
Lastly, all adolescents participating in the study underwent radiography. Naturally, doing radiography for research can be problematic. This study only targeted people who met the following conditions: 1) People who had not taken radiological images for the purpose of examination or treatment within one year; 2) People who were suspected of having a disease such as turtle neck (FHP) on the outside but could not be tested; People who want a clear radiology test for FHP, 3) People who have heard a sufficient explanation about the biological problems that radiography may cause and have given consent. And explanation and consent for all of the above items were provided to each target child and guardian. Moreover, only minimal images (one lateral spine image taken while standing) were taken for X-ray radiography of voluntary participants who met the three conditions mentioned above and sought a diagnosis of FHP. By progressing this study, the FHP screening test method will be developed to minimize unnecessary radiological examinations for growing children in the future, and guidelines for radiographic imaging will be developed. This study was conducted because radiography was necessary for the validity of the study.
- Improve the figure quality: Improve the figure quality of your manuscript especially figure no. 3
[Authors’ response] Thank you. According to your comment, the figures have been replaced with a high-quality version. We prepared and individually separated the figure with the 1447 × 2582 pixel size as 300 dpi. If the figure’s quality is still poor when you check, it can be a correct effect by the MS Office. In that case, please check the separated figures.
[Modified Figure, Page: 8, Line: 286]
[Modified Figure, Page: 9, Line: 297]
- Comparison with other state-of-art approaches: Make a comparison with other pre-existing approaches.
[Authors’ response] Thank you for your good comment. We have found the state-of-art related to this study. Hilla et al. (2023) investigated the relationship between FHP and neck pain, employing craniovertebral angle measurements and specialized virtual reality software. Their findings indicated no significant correlation between FHP and neck pain, offering a contrasting perspective to our approach which combines noninvasive measurements with radiographic data. Chu et al (2020) explored the impact of FHP on upper cervical spine stability. This study utilized radiographic analysis to assess changes in joint spacing related to FHP, offering insights into its potential effects on cervical spine stability. In contrast, our research focuses on a diagnostic model that integrates both noninvasive body angle measurements and detailed radiographic parameters. Our study stands out by providing a comprehensive diagnostic model for FHP, and integrating both noninvasive body angle measurements and detailed radiographic parameters.
[Added Contents, Page: 15, Line: 474-484]
Sarig Bahat et al. (2023) investigated the relationship between FHP and neck pain, employing craniovertebral angle measurements and specialized virtual reality software. Their findings indicated no significant correlation between FHP and neck pain, offering a contrasting perspective to our approach which combines noninvasive measurements with radiographic data. Chu et al (2020) explored the impact of FHP on upper cervical spine stability. This study utilized radiographic analysis to assess changes in joint spacing related to FHP, offering insights into its potential effects on cervical spine stability. In contrast, our research focuses on a diagnostic model that integrates both noninvasive body angle measurements and detailed radiographic parameters. Our study stands out by providing a comprehensive diagnostic model for FHP, and integrating both noninvasive body angle measurements and detailed radiographic parameters.
[Added References, Page: 17, Line: 600-603]
28) Sarig Bahat, H.; Levy, A.; Yona, T. The association between forward head posture and non-specific neck pain: A cross-sectional study. Physiother Theory Pract 2023, 39 1736–1745. DOI: 10.1080/09593985.2022.2044420
29) Chu, E.C.; Lo, F.S.; Bhaumik, A.; Plausible impact of forward head posture on upper cervical spine stability. J Family Med Prim Care, 2020, 9, 2517–2520. DOI: 10.4103/jfmpc.jfmpc_95_20.
- Contribution of the work: Write a proper way of mentioning the contribution of your work before the end of the introduction section. Authors have mentioned diagnostic models, it is requested you to mention which diagnostic models are applicable in your research.
[Authors’ response] Thank you for your insightful suggestion. In accordance with your suggestion, we have revised the introduction section of our manuscript to explicitly state the contribution of our work.
[Added Contents, Page: 3, Line: 102-112]
This study introduces a preliminary diagnostic model for FHP in adolescents, which combines the use of a noninvasive body angle measurement system, specifically assessing the FNTA with detailed radiographic analysis. This preliminary model enhances existing diagnostic approaches by providing a more comprehensive assessment of FHP including its potential impact on the cervical spine. Especially, the approach in this study offers a significant advancement over current models by integrating both noninvasive and radiographic techniques. This dual methodology not only allows for preliminary noninvasive screening but also ensures detailed diagnostic accuracy through radiographic evaluation. This is particularly crucial in the adolescent population, where early detection and accurate diagnosis of FHP are essential for effective intervention and prevention of long-term complications.
Reviewer 3 Report
Comments and Suggestions for Authors
Excellent work, with a correct design. I would like to make some remarks to try, if the authors consider it, to improve the manuscript:
- I think it should be reflected how the recruitment of patients has been. We do not know if it has been sequential, if there has been some kind of sample size calculation, etc.
- Given the sample size, why is the Kolmogorov-Smirnov test not performed?
- Are the same professionals performing the different measurements?
- Have the authors considered performing an ROC curve?
- Is it feasible to perform an Intraclass Correlation Coefficient?
For the rest, the authors can only be congratulated.
Author Response
Reviewer 3
Excellent work, with a correct design. I would like to make some remarks to try, if the authors consider it, to improve the manuscript:
[Authors’ response] Thank you for consuming your precious time to improve the quality of our manuscript with your great comments. And we really appreciate to bring the positive aspects of our study to us. According to your comments, our manuscript has been revised. We would like to ask you to review our responses for your comments.
- I think it should be reflected how the recruitment of patients has been. We do not know if it has been sequential, if there has been some kind of sample size calculation, etc.
[Authors’ response] Really appreciate. All children participating in the study underwent radiography. Naturally, doing radiography for research can be problematic. Accordingly, our researchers only targeted people who met the following conditions: 1) People who had not taken radiological images for the purpose of examination or treatment within one year; 2) People who were suspected of having a disease such as turtle neck (FHP) on the outside but could not be tested; People who want a clear radiology test for FHP, 3) People who have heard a sufficient explanation about the biological problems that radiography may cause and have given consent. In addition, explanation and consent for all of the above items were provided to each target child and guardian. In addition, only minimal images (one lateral spine image taken while standing) were taken for X-ray radiography of voluntary participants who met the three conditions mentioned above and sought a diagnosis of FHP. The impact of one plain X-ray radiography on the human body is minimal. In addition, among the various radiographs taken to check the condition of the spine, side imaging of the spine has a smaller effect than anterior and posterior imaging. Also, please consider that the study volunteers wanted to diagnose FHP, and radiography was performed for diagnostic purposes. An additional discussion has been added into the manuscript.
[Added Contents, Page: 14, Line: 459-473]
Lastly, all adolescents participating in the study underwent radiography. Naturally, doing radiography for research can be problematic. This study only targeted people who met the following conditions: 1) People who had not taken radiological images for the purpose of examination or treatment within one year; 2) People who were suspected of having a disease such as turtle neck (FHP) on the outside but could not be tested; People who want a clear radiology test for FHP, 3) People who have heard a sufficient explanation about the biological problems that radiography may cause and have given consent. And explanation and consent for all of the above items were provided to each target child and guardian. Moreover, only minimal images (one lateral spine image taken while standing) were taken for X-ray radiography of voluntary participants who met the three conditions mentioned above and sought a diagnosis of FHP. By progressing this study, the FHP screening test method will be developed to minimize unnecessary radiological examinations for growing children in the future, and guidelines for radiographic imaging will be developed. This study was conducted because radiography was necessary for the validity of the study.
- Given the sample size, why is the Kolmogorov-Smirnov test not performed?
[Authors’ response] Thank you for your valuable question. Actually, we considered two normality tests as the Shapiro–Wilk normality test and the Kolmogorov-Smirnov test. To our best knowledge, the Kolmogorov-Smirnov test is proper for the thousands of data samples. Whereas, the small amount of data shows good results when the normality test is conducted using Shapiro–Wilk normality test. For example, the statistical analysis software SAS from SAS Institute automatically performs the normality test as the Shapiro–Wilk normality test when the sample size is under 2,000. That is why we used the Shapiro–Wilk normality test instead of the Kolmogorov-Smirnov test.
- Are the same professionals performing the different measurements?
[Authors’ response] Thank you for your meaningful question. To keep the consistency of the measurement, one specialist conducted the measurement for all data. And another specialist verified all measurement results with the same method.
- Have the authors considered performing an ROC curve?
[Authors’ response] Thank you for your considerate question. In the context of this study, the ROC curve may not be necessary for Figure 5 if the aim is to assess diagnostic accuracy through other means. A confusion chart, like the one used in Figure 5, can provide a direct comparison of true and false, positives and negatives, which might be more relevant for your analysis. ROC curves are typically used to assess the diagnostic ability of a binary classifier system, However, if the study is focused on comparing multiple diagnostic models using specific metrics like TPR, FNR, PPV, and FDR, then a confusion chart might be more appropriate and informative for the dedicated research objectives. This approach can effectively highlight the strengths and weaknesses of each diagnostic model in a more straightforward manner for this study. For this reason, we have employed the confusion chart instead of the ROC curve.
- Is it feasible to perform an Intraclass Correlation Coefficient?
[Authors’ response] Thank you for your good question. We have considered several correlation models at the stage of the research design. Naturally, ICC was also one of the considerations for the statistical analysis in this study. However, we decided to use the statistical analysis model as the Spearman rho and Pearson’s correlation. To our best knowledge, the ICC is the test method to evaluate the repeatability, and reproducibility by several observers. Because all measurements were performed by one specialist and just verified by another specialist, we believed the ICC was not effective. Moreover, our methodology was focused on the finding of the correlation among several groups individually measured by one person, not finding the repeatability and reproducibility. For this reason, we did not employ the ICC as a statistical analysis in this study.
For the rest, the authors can only be congratulated.
[Authors’ response] We really appreciate your great and positive review. We could learn about a lot of things from your review comments. Thank you again.
Round 2
Reviewer 1 Report
Comments and Suggestions for Authors
The article can be accepted.
Author Response
The article can be accepted.
[Authors' Response] Really Appreciate. It was an honor to get your review to suggest good comments. Thanks again.